# Influence of Regular Physical Activity on Mitochondrial Activity and Symptoms of Burnout—An Interventional Pilot Study

**DOI:** 10.3390/jcm9030667

**Published:** 2020-03-02

**Authors:** Serge Brand, Katarina Ebner, Thorsten Mikoteit, Imane Lejri, Markus Gerber, Johannes Beck, Edith Holsboer-Trachsler, Anne Eckert

**Affiliations:** 1Psychiatric Clinics (UPK), Center for Affective, Stress and Sleep Disorders, University of Basel, 4001 Basel, Switzerland; thorsten.mikoteit@spital.so.ch (T.M.); edith.holsboer@gmail.com (E.H.-T.); 2Department of Sport, Exercise and Health, Division of Sport Science and Psychosocial Health, University of Basel, 4052 Basel, Switzerland; markus.gerber@unibas.ch; 3Substance Abuse Prevention Research Center, Health Institute, Kermanshah University of Medical Sciences (KUMS), 6719851115 Kermanshah, Iran; 4Sleep Disorders Research Center, Kermanshah University of Medical Sciences (KUMS), 6719851115 Kermanshah, Iran; 5School of Medicine, Tehran University of Medical Sciences, 1416753955 Tehran, Iran; 6Transfaculty Research Platform, Molecular and Cognitive Neuroscience, Neurobiology Laboratory for Brain Aging and Mental Health, Psychiatric Clinics (UPK), University of Basel, 4001 Basel, Switzerland; katarinaalexandra.ebner@usb.ch (K.E.); imane.lejri@upk.ch (I.L.); anne.eckert@upk.ch (A.E.); 7Psychiatric Services Solothurn and Faculty of Medicine of the University of Basel, 4503 Solothurn, Switzerland; 8Psychiatric Hospital Sonnenhalde, 4125 Riehen, Switzerland; johannes.beck@sonnenhalde.ch

**Keywords:** mitochondrial activity, physical activity, burnout, ATP, OXPHOS, decylubiquinone (DBQ)/hexaammineruthenium (HAR)-ratio, respiration

## Abstract

Background: Occupational burnout is both a serious public and individual health concern. Psychopharmacological and psychological interventions are often employed, while interventions involving physical activity have been less frequently studied. The aims of the present study were (1) to investigate the effects of physical activity on mitochondrial activity levels and symptoms of burnout, (2) to compare the mitochondrial activity levels and symptoms of burnout of individuals suffering burnout with those of healthy controls (HCs), and (3) to explore the associations between mitochondrial activity and burnout symptoms. Methods: Twelve males with burnout (mean age: M = 45.8 years) took part in the study. At baseline and after 12 weeks of an intervention involving physical activity, participants completed questionnaires covering symptoms of burnout and depression. In parallel, blood samples were taken to measure changes in mitochondrial functional outcomes, such as ATP levels, oxygen consumption and complex I. For comparison, baseline values of healthy controls (HCs; depression and burnout questionnaires; blood samples) were assessed. Results: Over time, symptoms of burnout (emotional exhaustion and depersonalization) and depression significantly decreased in participants with burnout (large effect sizes) but remained significantly higher than those of HCs (medium to large effect sizes). Personal accomplishment increased over time (medium effect size) but was still lower than for HCs (large effect size). At baseline and compared to HCs, individuals with burnout had significantly lower ATP levels of mitochondrial functional outcomes. Over time, mitochondrial activity levels increased among individuals with burnout. High baseline mitochondrial activity was significantly correlated with lower depression and burnout scores both at baseline and at the end of the study. Conclusions: In individuals with burnout, regular physical activity had positive effects on mitochondrial activity and on symptoms of burnout and depression. However, when compared to healthy controls, full remission was not achieved.

## 1. Introduction

Burnout describes a cognitive-emotional and behavioral state of emotional exhaustion, depersonalization and lack of personal accomplishment. This state is the result of a long-lasting imbalance between the demands of performing and completing tasks (at work), and the personal resources available to meet these demands [1].

Following Maslach and Jackson [1], emotional exhaustion is described as lack of emotional and physical energy. Typically, this involves complaints about emotional exhaustion either after some hours at work, or as a permanent state. In addition, feelings of emotional exhaustion are associated with symptoms of anxiety, frustration, and continual tiredness in the evenings and weekends. Depersonalization (lack of empathy; cynicism) reflects a negative attitude lacking in empathy towards the social environment, and more specifically towards costumers, patients, students, peers and colleagues at work. The lack of personal accomplishment reflects the quality and quantity of (workplace-related) performance [2].

To explain the emergence and maintenance of burnout, Brand and Holsboer-Trachsler [2] outlined a theoretical and dynamic model combining personality traits and workplace-related conditions. The relevant personality traits include perfectionism, overidentification with work, a high sense of duty, but also low social integration and adaptation to peers. Workplace-related factors include ‘high demand and low control’ conditions, and adverse working conditions such as poor support from peers and supervisors. Importantly, the process of emotional exhaustion, depersonalization and lack of personal accomplishment is gradual and develops slowly, with the subtle changes being difficult to perceive. Brand and Holsboer-Trachsler [2] argued that their theoretical model allows for personality traits and workplace-related issues to interact in a bi-directional fashion.

Suffering from burnout has serious physical, psychological and occupational consequences [3]. While there is broad consensus that people with burnout do suffer and do need medical and psychological treatment, the debate continues as to whether burnout should be considered a specific and well-defined psychiatric disorder or instead an epiphenomenon of an existing psychiatric disorder, such as major depressive disorder (ICD-10: F33.xx), adjustment disorder (ICD-10: F43.xx) [4,5,6,7,8,9], or chronic fatigue syndrome (ICD-10: G93.3). Unsurprisingly, there are no conclusive diagnostic criteria [10] and, in the Classification of Diseases (ICD 10), “burnout” is classified under Z.73.0 as “Burnout syndrome (state of total exhaustion)”. Thus, on a psychiatric and affective level, individuals with burnout show similar traits to those with major depressive disorder and chronic fatigue.

To further clarify, if burnout is to be considered a psychiatric classification in its own right, robust biomarkers would be helpful. However, as regards the association between burnout and biomarkers, in their systematic review Danhof-Pont et al. [11] found that in 31 studies, 38 different biomarkers had been assessed focusing variably on the hypothalamus–pituitary–adrenal axis, autonomic nervous system, immune system, metabolic processes, antioxidant defense, hormones, and sleep. Accordingly, given the heterogeneity of outcome variables, along with methodological variations in defining and operationalizing symptoms of burnout, it is not surprising that no clear-cut picture linking biomarkers to burnout has yet been described. Traunmüller et al. [12] investigated middle-aged adults with either low or high risk of burnout and concluded that burnout might not necessarily imply physiological disturbances.

Nevertheless, here, we focused on mitochondrial activity, as this is key in explaining cell activity. When exposed to stress, a cell will activate a multitude of mechanisms to maintain homeostasis and, in order to meet changing energy requirements, mitochondria provide the cell with the energy currency adenosine triphosphate (ATP). Mitochondrial activity and its functional outcomes, such as ATP, are also up-regulated in situations of high energy demand such as physical activity [13,14,15,16].

By contrast, whereas the association between regular physical activity and mitochondrial function has been well documented (see below), this is not the case for the association between burnout traits and mitochondrial function. This is surprising for two reasons. First, the overlap between burnout traits and major depressive disorders [4,17,18,19,20] and fatigue [21] are well documented. Second, there is sufficient evidence that in people with major depressive disorders [22] and fatigue [23] mitochondrial activity is decreased. Specifically, Picard et al. [22] showed in their systematic review that compared to healthy controls, individuals with major depressive disorders had down-regulated mitochondrial activity. Likewise, Tomas and Newton [23] reported in their mini-review that compared to healthy controls, individuals with chronic fatigue syndrome had deteriorated mitochondrial dysfunction. Given this, it appeared plausible that also among individuals with burnout traits, mitochondrial activity should be dysfunctionally down-regulated.

In the present study, we investigated to what extent a program of regular physical activity would positively impact on markers reflecting mitochondrial activity such as ATP levels, the amount of oxygen consumed by the oxidative phosphorylation system (OXPHOS) and the activity as well as content of the complex I of the electron transport chain (ETC) in mitochondria.

While research has confirmed the beneficial effect of regular physical activity for individuals with major depressive disorders [24,25,26,27,28], the benefits are less clear with respect to burnout. In a previous study [29], we showed that, after a 12 week intervention involving physical activity, profiles of mood states improved and stress decreased. Likewise, Beck et al. [30] showed that, following a physical activity intervention, impaired cognitive performance normalized. de Vries et al. [31] showed in their intervention study that in the case of work-related fatigue, exercise does constitute a powerful medicine for those who comply with the treatment. In their systematic review, Naczenski et al. [32] analyzed ten intervention studies and concluded that physical activity appears to be an effective means of reducing symptoms of burnout, though the quality of the methodology was not consistently satisfactory. Lindegård et al. [33] showed that regular physical activity has the potential to make the effects of multimodal treatment of burnout patients more sustainable.

To summarize, programs of physical activity appear offer promising treatments of burnout, and one key marker might be mitochondrial activity. To test these possibilities, we investigated the burnout symptoms and mitochondrial activity of 12 males with burnout both at baseline and after a 12 week program of physical activity. We also compared their data with those from healthy controls. Beyond simply filling a gap in research, we believe that the present study has the potential to clarify the associations between physical activity, mitochondrial activity and mental states such as symptoms of depression. We also believe that the results could be important for individuals with burnout, because they suffer from a severe inability to rest or to feel fully functional both in professional and private life. Finally, the results are clinically important by virtue of the potential to provide further insight into the importance of mitochondrial activity during recovery from burnout and within a thorough physical activity intervention.

The following three hypotheses and one research question were formulated. First, following others [22,23], we expected that at baseline and compared to healthy controls, mitochondrial activity would be lower in individuals with burnout traits. Second, following others [13,14,15,16], we anticipated that mitochondrial activity would increase following a 12 week program of physical activity. Third, we expected that among individuals with burnout, increased mitochondrial activity would be associated with a reduction in burnout symptoms. Last, as an exploratory research question, we considered whether the mitochondrial activity levels and burnout symptoms of individuals with burnout would be comparable to those of healthy controls following the intervention.

## 2. Method

### 2.1. Procedure

As outlined elsewhere [29,30], males with self-reported symptoms of burnout were approached to participate in the study. The aims, study design, and anonymous data handling were fully explained. Thereafter, participants signed a written informed consent. At baseline (t0) and 12 weeks later (study end; t1), participants with burnout completed a series of burnout- and depression-related questionnaires. Blood samples were also taken at these two time points (t0; t1). The intervention consisted of two to three endurance and resistance sessions per week (see below).

A sample of healthy controls (HCs) was recruited to compare symptoms of depression and burnout and mitochondrial activity concentrations with those of participants suffering burnout. While HCs had a single assessment at baseline (t0), their data were compared with data of individuals with burnout both at baseline (t0) and at the end of the study (t1). The Ethical Committee of Basel (Ethikkommission beider Basel, Basel, Switzerland: EK: 22/08; January 22, 2008) approved the study, which was performed in accordance with the rules laid down in the Declaration of Helsinki.

### 2.2. Samples

#### 2.2.1. Participants with Burnout

As described in more detail elsewhere [22,23], the sample consisted of twelve males (M= 45.8 years, SD = 6.8; range 36–65 years) suffering from burnout syndrome, based on Maslach’s definition of occupational burnout. All participants continued working during their participation in the study. Inclusion criteria were: 1. Age between 18 and 65 years; 2. Male; 3. Score of 27 or higher on the Maslach Burnout Inventory (see below); 4. Compliance with the study conditions; 5. Signed written informed consent. Exclusion criteria were: 1. Tobacco use; 2. Somatic issues such as metabolic diseases, neurological issues, liver or renal dysfunction, acute or chronic infectious diseases, as ascertained by a brief medical interview; 3. Psychiatric issues such as substance use disorder, major depressive disorders, bipolar disorder, personality disorder, or post-traumatic stress disorder (PTSD), as ascertained by a brief psychiatric interview [34]; 4. Current musculoskeletal injury impeding regular physical activity, 5. Regular physical exercising; 6. Current psychotherapeutic or psychopharmacological treatment, including regular intake of omega-3-polyunsaturated fatty acids [35,36].

#### 2.2.2. Healthy Controls

A sample of healthy controls was assessed at baseline. Inclusion criteria were: 1. Age between 18 and 65 years; 2. Male; 3. Score of 5 points or lower on the Maslach Burnout Inventory (see below); 4. Compliance with the study conditions; 5. Signed written informed consent. Exclusion criteria were: 1. Tobacco use; 2. Somatic issues such as metabolic diseases, neurological issues, liver or renal dysfunction, acute or chronic infectious diseases, as ascertained by a brief medical interview; 3. Psychiatric issues such as substance use disorder, major depressive disorders, bipolar disorder, personality disorder, or PTSD, as ascertained by a brief psychiatric interview [34]; 4. Current musculoskeletal injury impeding regular physical activity; 5. Regular physical exercising; 6. Current psychotherapeutic or psychopharmacological treatment.

#### 2.2.3. Physical Activity Intervention

As extensively described elsewhere [29,30] and following Dunn et al. [37], the intervention lasted for 12 consecutive weeks and consisted of supervised and monitored sessions three times a week for 60 min. Participants were instructed to exercise to within 60–75% of their maximum heart rate. Heart rate was monitored during all training sessions with a chest belt heart rate monitor (Polar^®^) to ensure exercising below the anaerobic threshold. Participants were instructed that during the exercise intervention, they should engage only in any physical activities that they were performing on a regular basis prior to the beginning of the program (e.g., cycling to work).

### 2.3. Tools

#### 2.3.1. Sociodemographic Information

#### 2.3.2. Participants Reported Their Age

#### 2.3.3. Burnout Questionnaire

Participants completed the Maslach Burnout Inventory [1] which assesses symptoms of burnout. The questionnaire consists of 22 items providing scores for emotional exhaustion, depersonalization and (lack of) personal accomplishment. Typical items are “I feel burned out from my work.” (emotional exhaustion); “I don’t really care what happens to some other people.” (depersonalization/lack of empathy); “I have accomplished many worthwhile things in this job.” (personal accomplishment). Answers are given on seven-point rating scales, ranging from 0 (= never) to 6 (= every day), with higher scores reflecting higher impairments on the dimensions of emotional exhaustion and depersonalization. In contrast, higher scores on personal accomplishment reflect greater resources and workplace-related satisfaction.

#### 2.3.4. Depression

Participants completed the Beck Depression Inventory [38], which samples self-reported symptoms of depression. The questionnaire consists of 21 items and addresses several dimensions including depressive mood, loss of appetite, sleep disorders, and suicidality. Each question has a set of at least four possible responses covering a range of intensities—e.g., ‘sadness’: 0 = ‘I do not feel sad’; 1 = ‘I feel sad’; 2 = ‘I am sad all the time and I can’t snap out of it’; 3 = ‘I am so sad or unhappy that I can’t stand it.’—with higher scores reflecting greater severity of depressive symptoms.

#### 2.3.5. Biochemical Procedures and Mitochondrial Functional Assays

Mitochondrial activity was determined from blood samples drawn before and after the 12 week exercise intervention to measure changes in levels of mitochondrial functional readouts such as ATP levels, oxygen consumption, mitochondrial membrane mass and complex I activity and content in mitochondria by following four protocols previously described by Rhein et al. [32,33]. Mitochondrial oxygen consumption, ATP levels and estimation of mitochondrial membrane mass were determined in freshly isolated vital lymphocytes. Complex I enzyme activity and content were measured in isolated platelet mitochondria.

*ATP levels.* Total ATP content was determined using a bioluminescence assay (ViaLighTM HT, Cambrex Bio Science, Walkersville, MD, USA) according to the instruction of the manufacturer, as previously described [32]. Freshly isolated lymphocytes were plated at a density of 2 × 10^5^ cells per well in a white 96-well plate. The bioluminescent method utilizes the enzyme luciferase, which catalyzes the formation of a bioluminescent molecule from the substrates ATP and luciferin. ATP concentration is linearly related to the emitted light, which was measured using a luminometer [32].

*Mitochondrial respiration readouts*. Mitochondrial oxygen consumption was measured following the oxidative phosphorylation system (OXPHOS) [32], using an Oroboros Oxygraph-2k system at 37 °C. This high-resolution protocol enables the determination of oxygen consumption in whole, vital cells under most physiological conditions. Eight million lymphocytes were added to 2 mL of a mitochondrial respiration medium containing 0.5 mM EGTA, 3 mM MgCl2, 60 mM K-lactobionate, 20 mM taurine, 10 mM KH2P04, 20 mM HEPES, 110 mM sucrose, 1 g/L BSA (pH 7.1). To measure state 4 (= state 2) of complex I, 10 mM glutamate and 2 mM malate were added and cells were permeabilized with 40 µg/mL digitonin. Afterwards, 2 mM ADP was added to measure state 3. The integrity of mitochondrial membrane was checked through the addition of 10 µM cytochrome c. After determining coupled respiration, 0.4 µM FCCP (Carbonyl cyanide p-(trifluoromethoxy)phenyl-hydrazone) was added and respiration was measured in the absence of a proton gradient. In order to inhibit complex I and III activities, 0.5 µM rotenone and 2.5 µM antimycine A respectively were added. Then, 2 mM ascorbate and 0.5 mM TMPD were added to give access to complex IV activity. Finally, 100 mM sodium azide was added to inhibit mitochondrial respiration. To avoid oxygen limitation, an intermittent re-oxygenation was achieved through partial opening of the oxygraph chamber and oxygen transfer from the gas phase, until a maximum oxygen concentration was obtained.

*Amount of mitochondria.* Estimation of mitochondrial membrane mass that corresponds to the amount of mitochondria was performed as described previously [33,34]. Isolated lymphocytes were plated at a density of 1 × 10^6^ cells per well in a black 96-well plate. Mitochondrial membrane mass was measured using the cell-permeable mitochondria-selective dye MitoTracker Green FM (Molecular Probes, Invitrogen, Lucerne, Switzerland). This probe accumulates in active mitochondria independently of mitochondrial membrane potential and then reacts with accessible thiol groups of peptides. Fluorescence was determined using a Fluoroskan Ascent FL multiplate reader (Thermo Labsystems, Milford, MA, USA), exciting at 490 nm and measuring the emission at 516 nm.

*Mitochondrial complex I readouts*. Activity of complex I was directly measured in freshly isolated and solubilized platelet mitochondria. Decylubiquinone (DBQ) is a ubiquinone analogue serving as a substrate for NADH-ubiquinone oxidoreductase (NADH-DBQ activity). NADH-DBQ activity was normalized to complex I content (hexaammineruthenium (III) chloride (HAR)-reductase activity) and is referred to as the DBQ/HAR ratio. In brief, a total of 300 µg of isolated mitochondria was solubilized via sonification (20 sec). NADH:hexaammineruthenium(III)-chloride (HAR) activity was measured at 30 °C in a buffer containing 2 mM Na^+^/MOPS, 50 mM NaCl, and 2 mM KCN, pH 7.2, using 2 mM HAR and 200 µM NADH as substrates to estimate complex I content. To determine NADH-ubiquinone oxidoreductase activity, 100 µM n-decylubiquinone (DBQ) and 100 µM NADH were used as substrates and 5 µM rotenone as inhibitor, as described previously [32,34]. Oxidation rates of NADH were recorded with a Shimadzu Multi Spec-1501 diode array spectrophotometer (ε_340–400_ nm =6.1 mM^−1^ cm^−1^). Complex I activity was in addition normalized to the complex I content of the mitochondrial preparation and is given as the DBQ/HAR ratio.

#### 2.3.6. Statistical Analysis

### 2.4. The Following Were the Key Outcome Variables

Symptoms of burnout: Emotional exhaustion; depersonalization; personal accomplishment (Maslach Burnout Inventory).Symptoms of depression: Beck depression scores.Mitochondrial activity: ATP; CI_DBQ; CI_HAR; DBQ/HAR ratio; O_2_ consumption state 1/4/3/3u; ascorbate/TMPD (A/T). Note that at baseline, ATP levels as well as CI_DBQ and CI_HAR were available for samples both with and without burnout; mitochondrial respiration data were only available for the sample with burnout.

To compare sociodemographic data of participants with and without burnout, X^2^-tests and a *t*-test were performed.

A series of *t*-tests for related samples was performed to test for changes in burnout and depression scores and mitochondrial activity levels from baseline (t0) to the study end (t1).

To compare burnout and depression scores and mitochondrial activity concentrations between individuals with and without burnout, a series of *t*-tests was performed according to the following strategy:

*At baseline (t0)*: Symptoms of depression and burnout, and mitochondrial activity between individuals with burnout and healthy controls.

*At study end (t1)*: Symptoms of depression and burnout, and mitochondrial activity between individuals with burnout *at study end (t1)* and healthy controls *at baseline (t0).*

Pearson’s correlations were computed to test for associations between burnout and depression scores and mitochondrial activity concentrations in individuals with burnout. 

Effect sizes for *t*-tests were reported with Cohen’s d: sizes were evaluated as follows: ds 0–0.19: trivial effect size; ds 0.20–0.49: small effect size; ds: 0.50–0.79: medium effect size; ds: 0.80 and greater: large effect size.

The nominal level of significance was set at alpha < 0.05. All statistical computations were performed with SPSS^®^ 25.0 (IBM Corporation, Armonk NY, USA) for Apple Mac^®^.

## 3. Results

### 3.1. Samples

There was no age difference between individuals with burnout (M = 45.80 years; SD = 6.84) and those without (M = 45.65 years; SD = 6.86; t(22) = 0.07, *p* = 0.95, d = 0.01).

### 3.2. Symptoms of Burnout and Depression in Individuals with Burnout and Healthy Controls; Baseline and Study End

Table 1 reports the descriptive and inferential statistical indices of symptoms of burnout and depression separately for the two time points (baseline (t0); study end (t1)), and for individuals with burnout and healthy controls.

At baseline and compared to healthy controls, individuals with burnout had more severe symptoms of depression, higher scores for emotional exhaustion and depersonalization, and lower scores for personal accomplishment (in all cases, large effect sizes).

In individuals with burnout and following the intervention symptoms of depression, emotional exhaustion, and depersonalization decreased (in all cases, large effect sizes), and personal accomplishment increased (medium effect size).

At the end of the study (t1), and compared to values for healthy control *at baseline (t0)*, the symptoms of depression, emotional exhaustion, and depersonalization of individuals with burnout were higher (medium to large effect sizes), and personal accomplishment scores were lower (large effect size).

To summarize, in individuals with burnout, and after an intervention involving regular physical activity, symptoms of depression and burnout improved, though scores did not reach the level of those of healthy controls.

### 3.3. ATP Level in Individuals with Burnout and Impact of Physical Intervention

At baseline (t0), ATP was available for individuals with and without burnout. At study end (t1), ATP was available for individuals with burnout. As for symptoms of depression and burnout, the baseline ATP values of healthy controls were compared with the ATP values of individuals with burnout both at baseline (t0) and at study end (t1).

Table 2 and Figure 1A give the descriptive and inferential statistical indices of ATP levels in individuals with burnout (baseline and study end) and without burnout (baseline).

At baseline (t0), ATP levels were significantly lower in individuals with burnout than in healthy controls (Figure 1A; Table 2, large effect size), while mitochondrial membrane mass did not differ between the groups (data not shown), suggesting that the deficit in ATP is not associated with reduced mitochondrial numbers.

In individuals with burnout, ATP levels significantly increased from baseline to the study end (Figure 1A; Table 2, large effect size), but compared to ATP concentrations of healthy controls *at baseline*, ATP levels of individuals with burnout were still lower at *study end (t1)* (Table 2, medium effect size).

To test whether the reduced ATP level in individuals with burnout was associated with a mitochondrial complex dysfunction, we analyzed mitochondrial enzyme activity and content of complex I, the largest and first enzyme of the electron transport chain (Figure 2). At baseline, CI_HAR activity, a measure of the mitochondrial complex I protein content, was significantly lower in individuals with burnout than in healthy controls (t0; Figure 2A). CI_DBQ activity—corresponding to the “genuine” complex I activity—was also significantly lower at baseline in individuals with burnout than in healthy controls (t0; Figure 2B). Consistent with this, a significantly reduced DBQ/HAR ratio was found in individuals with burnout (Figure 2C) indicating a burnout-related “functional” reduction of complex I activity in addition to an already reduced complex I protein content in burnout.

At study end (t1), a significantly increased mitochondrial complex I activity (CI_DBQ as well as the DBQ/HAR ratio) was observed in individuals with burnout (Figure 2B,C and Table 3, large effect size) but not in complex I content, which showed only a trend (CI_HAR; Figure 2A and Table 3, trivial effect size).

Because complex I is the major entry point for electrons to the electron transport chain and is suggested as the rate-limiting step in overall respiration of mitochondria, we assessed whether the improvement of complex I deficiency in individuals with burnout after physical intervention (t1) corresponded to an amelioration in oxygen consumption. Thus, we determined the oxygen consumption rates in burnout mitochondria at baseline (t0) and at study end (t1). The state 3 and 4 respiration of burnout mitochondria was detected by using substrate for complex I (glutamate/malate) (Figure 3). State 3 respiration measures the capacity of mitochondria to metabolize oxygen and the selected substrate in the presence of a limited quantity of ADP, which is a substrate for the ATP synthase (complex V). State 4 respiration measures respiration when all ADP is exhausted, and it is associated with proton leakage across the inner mitochondrial membrane. Therefore, it represents a “basal-coupled” rate of respiration.

While state 1, the “genuine” oxygen consumption rate, and state 4 respiration remained unchanged between baseline (t0) and the study end (t1), a significantly increased state 3 oxygen consumption rate was observed (Figure 3; Table 3, large effect size). In addition, after uncoupling with FCCP, the oxygen consumption rate state 3u in the absence of a proton gradient had significantly increased by the study end (Figure 3; Table 3, large effect size), indicating an improved maximum capacity of the electron transport chain for individuals with burnout after a program of physical activity. In the presence of the final respiratory complex IV substrate ascorbate and TMPD (A/T), the oxygen consumption rate A/T had also improved following the intervention (Figure 3; Table 3, large effect size).

Taken together, respiratory analyses showed that the intervention enhanced metabolic pathways in burnout mitochondria by increasing the overall oxygen consumption rate in coupling as well as maximum capacity, which could be due to the ameliorated complex I activity of burnout mitochondria. These changes may also explain the observed increase in ATP levels in burnout mitochondria at study end (t1).

### 3.4. Correlations between Burnout and Depression Scores and Mitochondrial Activity

Table 4 reports correlation coefficients between burnout and depression scores and mitochondrial activity readouts at baseline (t0) and at the end of the study (t1). 

At baseline, higher ATP levels and complex I content (CI_HAR) were associated with lower depression scores. Additionally, higher ATP levels and complex I content (CI_HAR) at baseline predicted lower scores of depression and emotional exhaustion. The relation between ATP levels and emotional exhaustion is also shown for the whole group at baseline in Figure 1B.

Next, higher O_2_ consumption state 3 levels at baseline were associated with higher scores for depression and emotional exhaustion, both at baseline and at the end of the study.

For all other mitochondrial activity levels and dimensions of depression and burnout, there were no descriptive or statistically significant correlations.

## 4. Discussion

The key findings of the present study were that, in a sample of males with burnout, intervention involving physical activity and lasting 12 weeks reduced symptoms of burnout and depression and improved mitochondrial activity. However, at the end of the intervention, levels of burnout and depression and mitochondrial activity outcomes were still lower than those of healthy controls. Furthermore, higher mitochondrial activity at baseline predicted lower burnout and depression scores at the end of the study. The present findings expand upon previous research in showing that, after an intervention involving physical activity, improvements with respect to burnout and depression were associated with improved dimensions of mitochondrial activity outcomes.

Three hypotheses and one research question were formulated and each of these is considered in turn.

With the first hypothesis, we expected that at baseline and compared to healthy controls, mitochondrial activity would be lower in individuals with burnout traits, and data did confirm this assumption. Accordingly, the pattern of results is in line with findings in individuals with major depressive disorders [22] and fatigue [23]. However, the present results expand upon previous research in that we showed a deteriorated mitochondrial activity among individuals with burnout. It further follows that the present study adds to the current literature in an important and novel way: major depressive disorders, fatigue and burnout do not only share common features at symptom level, but also at cellular level.

Our second hypothesis was that in individuals with burnout, and following a program of physical activity, mitochondrial activity parameters would increase, and this was supported (see Table 2 and Table 3). Accordingly, the present results are in line with previous findings [13,14,15,16]. However, the present results expand upon previous research in that this pattern of results was found among individuals with burnout, and after 12 weeks of endurance exercising two to three times a week.

Our third hypothesis was that improvements in burnout and depression and mitochondrial activity outcomes would be associated, but this was not fully supported (see Table 4). Higher CI_HAR levels at baseline (t0) were associated with lower scores for depersonalization at baseline (t0) and at the end of the study (t1), suggesting that, compared to other mitochondrial activity dimensions, CI_HAR may be the most appropriate predictor of psychopathological dimensions. In contrast, higher scores for depression and emotional exhaustion at baseline predicted higher O_2_ consumption state 3 at the end of the study. In our opinion, this pattern of results is counter-intuitive, though the evidence from the study cannot fully explain this pattern. Inspection of the scatter plot also revealed no outliers which might otherwise have biased the pattern of association. To summarize, we believe that improvements at the cellular level are not entirely mirrored in corresponding improvements at the behavioral level.

The exploratory research question was that, following the intervention described above, symptoms of burnout and mitochondrial activity levels in individuals initially diagnosed as suffering from burnout would be comparable to those of healthy controls, but this was not supported. As shown in Table 1 and Table 2, after the intervention, symptoms of burnout and depression decreased in individuals with burnout, though their burnout and depression scores nevertheless remained higher than those of healthy controls. Accordingly, it appears that after 12 weeks of physical activity, symptoms of burnout and depression do clearly reduce, but individuals with burnout do not fully recover. The data available from the study are unable to shed any direct light on the underlying neurophysiological mechanisms. In the absence of such direct evidence, we offer the following possibilities. First, by definition, burnout is a slow process lasting several months from the first signs of emotional exhaustion, depersonalization and lack of personal accomplishment to full-blown symptoms and sick leave [2,6,9,19]. It therefore seems very unlikely that full recovery could occur within the span of just 12 weeks. Second, Merrill et al. [39] for example, in order to improve sleep, coping and depression, employed an intensive *daily*, lifestyle intervention program lasting four weeks and consisting of ten or more hours of moderate to vigorous exercising, along with psycho-education and counselling on stress management. While the sample in Merrill et al.’s study (2624 adults aged from 30 to 80 years) is not comparable to the present sample of males with burnout, we also note that both the quantity (high intensity; high frequency; long duration) and the quality (physical activity; psychoeducation; counseling) of the intervention described in Merrill et al. [39] appeared to be more successful than a 12 week intervention involving moderate physical activity three times a week. Third, Brand and Holsboer [2] noted in their overview that the emergence and maintenance of burnout appears to be a complex process of environmental demands and (dysfunctional) personality traits. If this is the case, it is unlikely that dysfunctional personality traits would have been modified by physical activity alone over a 12 weeks period. In a related vein, an intervention program for individuals with Multiple Sclerosis involving regular physical activity produced changes in state variables such as depression and sleep quality, but not trait dimensions such as mental toughness or intolerance of uncertainty [40]. To put it in another way, it would be unwise to overstate the impact of regular physical activity on a broad range of psychological functioning.

As regards mitochondrial activity concentrations, improvements were observed (see Figure 1, Figure 2 and Figure 3 and Table 2 and Table 3), but ATP levels were still significantly lower than those of healthy controls.

Despite the novelty of the findings, several limitations warn against their overgeneralization. First, the sample size was small, and a larger sample might have yielded other important results. However, we focused on effect size characteristics given the growing criticism of “significant *p*-values” [41]. Further, we followed Julious [42], who proposes a sample size of at least 12 participants per group for studies with more experimental character. Second, as regards symptoms of burnout and depression, we relied on self-ratings, while experts’ ratings might have enhanced the reliability of the results. Third, it is conceivable that further latent but unassessed physiological and psychological dimensions might have biased two or more variables in the same or opposite directions. Specifically, there exists literature showing that regular physical activity has a positive effect on sleep quality, which in turn improves mood [43,44,45]. Fourth, it remains unclear whether or to what extent the present findings might apply to females suffering burnout. Fifth, other control samples, e.g., individuals with burnout and treated with psychotherapy, medications, or counseling sessions but without involving bona fide interventions, might have allowed an estimation of the relative power of physical activity compared to established treatment algorithms. Sixth, a follow-up might have shed more light on the long-term impact of this type of intervention.

## 5. Conclusions

The present study consists of the following novelties: first, similar to individuals with major depressive disorders and fatigue, we were able to show that individuals with burnout had a dysfunctional mitochondrial activity; second, in individuals with burnout, physical activity interventions lead to an up-regulation of mitochondrial activity; third, compared to healthy controls, in individuals with burnout, after physical activity interventions of 12 weeks full remission could not be achieved, both at a behavioral and at an cellular level. Larger-scale studies should seek to replicate and confirm this pattern of results. Further, future studies should systematically test which physical activity interventions (type; duration; frequency; intensity) might lead to a full remission of burnout.

## Figures and Tables

**Figure 1 jcm-09-00667-f001:**
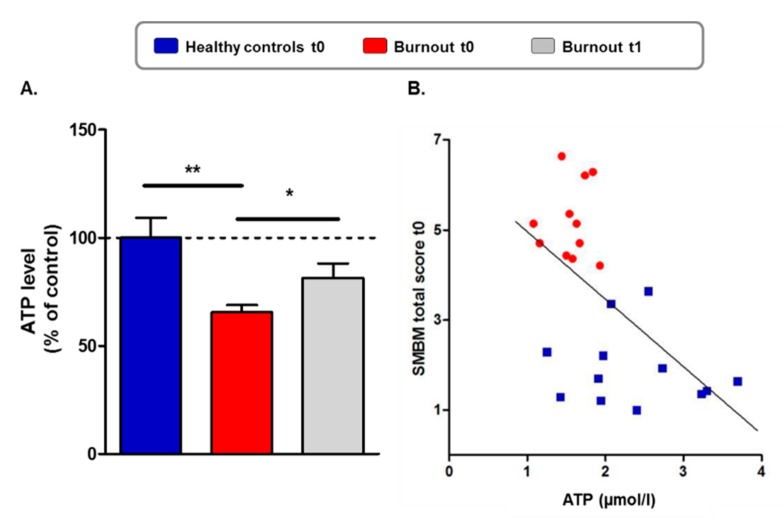
Reduced ATP levels in individuals with burnout and impact of physical activity. (**A**) ATP levels were normalized to the mean of the healthy control group at baseline (100%). ATP levels in the group of individuals with burnout at baseline (t0) were significantly lower compared to those of the healthy control group. At study end (t1), ATP levels significantly increased in individuals with burnout compared to baseline (t0). Student’s *t*-test * *p* < 0.05, ** *p* < 0.01. Values are represented by the mean ± SEM. (**B**) Correlation of absolute ATP levels (µmol/l per 2 × 105 cells) with total score of the Maslach Burnout Inventory (MBI) for the entire sample at baseline (t0). ATP levels decreased with increasing symptoms of emotional exhaustion. Pearson’s correlation coefficient −0.62 *p* < 0.001.

**Figure 2 jcm-09-00667-f002:**
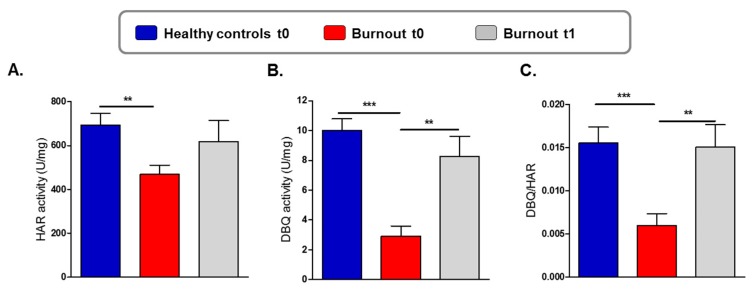
Reduced complex I content and activity in mitochondria from individuals with burnout and impact of physical activity. (**A**) Reduced complex I content as revealed by a significant reduction of NADH:HAR activity («HAR activity») in the group with burnout compared to healthy controls at baseline (t0). Student’s t test ** *p* = 0.0036. (**B**) Reduced “genuine” complex I activity as revealed by a significant reduction of NADH-ubiquinoneoxidoreductase (NADH:DBQ) activity («decylubiquinone (DBQ) activity) in the group with burnout compared to healthy controls at baseline (t0). Physical activity significantly increased DBQ activity in the burnout group at study end (t1). Student’s t test ** *p* = 0.0017, *** *p* < 0.0001. (**C**) The DBQ/HAR ratio was determined via normalization of complex I activity to the complex I content of the mitochondrial protein preparation. A significantly reduced DBQ/HAR ratio in individuals with burnout compared to healthy controls indicates a “functional” reduction of complex I activity at baseline (t0). Physical exercise improved significantly complex I activity in the burnout group. Student’s t test ** *p* = 0.0053, *** *p* = 0.0004. Values are represented by the mean ± SEM.

**Figure 3 jcm-09-00667-f003:**
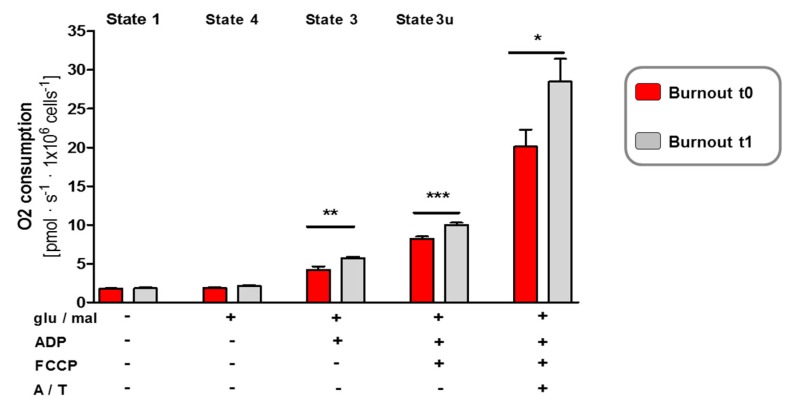
Beneficial effects of physical exercise on mitochondrial respiration in individuals with burnout. High-resolution respiratory measurement reveals an improvement in the mitochondrial oxidative phosphorylation system (OXPHOS) in mitochondria from individuals with burnout at study end. After detection of endogenous respiration (state 1), the mitochondrial substrates glutamate and malate (glu/mal) were added to induce state 4 respiration. ADP stimulated state 3 respiration. After determining coupled respiration, FCCP was added and the maximal respiratory capacity measured in the absence of a proton gradient. Complex IV activity was stimulated by ascorbate/TMPD (A/T). Two-way ANOVA revealed that the total cellular respiration (O_2_ consumption) significantly increased in the burnout group after physical activity compared to that at baseline before the intervention: effect of physical exercise *p* = 0.0015. Post-hoc Student’s *t*-test analysis: * *p* = 0.0355, ** *p* = 0.0046, *** *p* = 0.0005. Values are represented by the mean ± SEM.

**Table 1 jcm-09-00667-t001:** Descriptive and statistical indices of symptoms of burnout and depression in individuals with and without burnout.

		Groups				Statistical Comparisons			
	Baseline		Study end	Burnout		Burnout vs. HCs			
	Burnout	HC	Burnout	BL vs. SE		BL vs. BL		BL vs. SE	
N	12	12	12						
	M (SD)	M (SD)	M (SD)	t	d	t	d	t	d
Depression	17.58 (8.28)	4.25 (4.79)	7.42 (4.91)	t(11) = 4.72 **	1.49 (L)	t(22) = 4.83 ***	1.97 (L)	t(22) = 1.66	0.65 (M)
Burnout									
Emotional Exhaustion	40.17 (6.15)	4.71 (1.47)	26.75 (8.29)	t(11) = 4.75 **	1.48 (L)	t(22) = 19.42 ***	7.93 (L)	t(22) = 9.07 ***	3.70 (L)
Depersonalization	19.87 (4.67)	4.35 (1.67)	13.25 (5.16)	t(11) = 5.54 ***	1.35 (L)	t(22) = 10.95 ***	4.75 (L)	t(22) = 5.69 **	2.32 (L)
Personal Accomplishment	21.50 (5.48)	47.76 (1.69)	25.76 (7.47)	t(11) = 1.64	0.62 (M)	t(22) = 15.84 ***	6.48 (L)	t(22) = 10.05 ***	4.06 (L)

HC = healthy controls; BL = baseline; SE = study end; ** = *p* < 0.01; *** = *p* < 0.001. (S) = small effect size; (M) = medium effect size; (L) = large effect size.

**Table 2 jcm-09-00667-t002:** Descriptive and statistical indices of adenosine trisphosphate (ATP) concentrations in individuals with and without burnout; baseline and study end.

		Groups				Statistical Comparisons			
	Baseline		Study end	Burnout		Burnout vs. HCs			
	Burnout	HC	Burnout	BL vs. SE		BL vs. BL		BL vs. SE	
N	12	12	12						
	M (SD)	M (SD)	M (SD)	t	d	t	d	t	d
Adenosine Triphosphate	1.57 (0.27)	2.37 (0.76)	1.93 (0.49)	t(11) = 2.17	0.91 (L)	t(22) = 2.62 *	1.40 (L)	t(22) = 1.60	0.69 (M)

HCs = healthy controls; BL = baseline; SE = study end; * = *p* < 0.05. (S) = small effect size; (M) = medium effect size; (L) = large effect size.

**Table 3 jcm-09-00667-t003:** Descriptive and statistical indices of mitochondrial activity at baseline and study end in individuals with burnout.

	Time Points		Statistical Comparisons	
	Baseline	Study End		
N	12	12		
	M (SD)	M (SD)	t	d
C1_DBQ	2.02 (2.32)	8.28 (4.65)	t(11) = 4.18 **	1.71 (L)
C1_HAR	468.39 (146.96)	619.30 (327.10)	t(11) = 1.48	0.59 (M)
O_2_ state1	1.77 (0.26)	1.82 (0.50)	t(11) = 0.42	0.12 (T)
O_2_ consumption GM2	1.02 (0.50)	2.12 (0.41)	t(11) = 1.75	2.40 (L)
O_2_ consumption GM3	4.22 (1.49)	5.70 (0.69)	t(11) = 2.88 *	1.27 (L)
O_2_ consumption GM3u	8.19 (1.10)	9.99 (1.05)	t(11) = 5.23 ***	1.67 (L)
O_2_ consumption AT	20.09 (7.31)	29.56 (9.95)	t(11) = 3.36 *	1.08 (L)

HCs = healthy controls; BL = baseline; SE = study end; * = *p* < 0.05; ** = *p* < 0.01; *** = *p* < 0.001. (T) = trivial effect size; (M) = medium effect size; (L) = large effect size.

**Table 4 jcm-09-00667-t004:** Correlation coefficients between symptoms of depression and burnout and mitochondrial activity levels in individuals with burnout.

		Baseline			Study End	
	Depression	Emotional Exhaustion	Depersonalization	Depression	Emotional Exhaustion	Depersonalization
Baseline						
ATP	0.37	0.19	−0.29	−0.06	−0.59 *	−0.42
C1_DBQ	0.02	0.14	−0.23	−0.18	−0.27	−0.41
C1_HAR	0.08	−0.16	−0.86 **	−0.02	−0.61 *	−0.60 *
O_2_ Consumption State 1	0.06	−0.03	0.02	0.02	−0.09	0.31
O_2_ Consumption GM2	0.14	0.19	0.43	0.18	0.06	0.37
O_2_ Consumption GM3	0.05	0.10	0.49	0.10	0.08	0.29
O_2_ Consumption GM3u	−0.04	−0.09	0.54	0.13	0.38	0.54
O_2_ Consumption AT	−09	−0.04	0.09	−0.16	−0.29	−0.14
Study End						
ATP	0.30	0.40	−0.03	0.22	0.23	−0.15
C1_DBQ	0.10	0.12	−0.44	0.41	−0.11	−0.27
C1_HAR	−0.05	−0.03	−0.06	0.18	0.16	−0.19
O_2_ Consumption state 1	−0.19	−0.29	−0.34	−0.07	0.15	0.38
O_2_ Consumption GM2	0.14	−0.02	−0.21	−0.16	−0.06	0.29
O_2_ Consumption GM3	0.62 *	0.72 *	0.21	0.67 *	0.32	0.17
O_2_ Consumption GM3u	0.36	0.34	0.45	0.42	0.29	0.42
O_2_ Consumption AT	−0.08	−0.24	0.56 *	0.42	0.29	0.42

* = *p* < 0.05; ** = *p* < 0.01; *** = *p* < 0.001.

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
