# Peer review of "Influence of Regular Physical Activity on Mitochondrial Activity and Symptoms of Burnout—An Interventional Pilot Study"

_jcm, 2020, doi:10.3390/jcm9030667_

Round 1

Reviewer 1 Report

It was a pleasure to read this paper 

Influence of regular physical activity on mitochondrial activity and symptoms of burnout – an interventional pilot study 

This is a novel and important study that deserves publication.  The introduction is well written with these two concerns:

The case to study mitochondria in these patients with burnout could be better supported.  They offer good support that burnout is similar to MDD and chronic fatigue.  The connection between exercise and mitochondrial function was clearly made (13-16).  I believe the authors need linking support for their interest in studying mitochondria in people with burnout.  Since there is none, they can draw from data about decreased energy, depression, or increased fatigue.  this is a good review: Picard M, McEwen BS, Epel ES, Sandi C. An energetic view of stress: Focus on mitochondria. Front Neuroendocrinol. 2018 Apr;49:72-85. doi:10.1016/j.yfrne.2018.01.001. Epub 2018 Jan 12. Review.

2. The authors clearly stated the first and third hypotheses but their 2nd hypothesis was poorly stated in the introduction and it may need rewriting.  Along with this the authors may want to clearly state the other novel hypothesis which is that they expected that burnout phenomenology to be associated with reduced mitochondrial activity.  This sounds simple but I believe this in itself is a novel question and finding.

The inclusion and exclusion criteria were clearly stated and the measures were described very well.

Table 1 should include the age and a simple review of all the variables in the study at BL.  Table 2 could be devoted to the analysis such as comparisons of BL and SE.  

in my opinion the results and discussion should lead with the difference of the cases and controls at BL and only then proceed to focus on the hypotheses that the cases improved on psychological and physiological parameters with exercise.  this is a special population of burnout patients which has not been previously characterized in this way.

Author Response

Please see the detailed point-by-point-response and the revised manuscript. Thank you very much for all your kind efforts. 

Reviewer 2 Report

I found this manuscript to report on a particularly interesting and novel study. The investigators examine an intervention for which physical activity is a key component. This is of particularly note since physical activity interventions may be especially useful in terms of uptake, with very few side effects compared to many other interventions. The study is very well written and clear. The multi-method assessment is of particular note, as rating scales were collected in addition to blood samples that allowed an examination of mitochondrial activity levels. This is a very unique aspect of the study and the authors are to be commended for included biological assessment within a pilot study. The tables and figures are clear. Although full remission was not achieved in this study, the findings are nevertheless quite compelling and point to very important and interesting directions for future research. I personally was unable to find any errors or flaws in the study, and I think it will make a very good contribution to the literature. I thank the authors for the opportunity to review their very interesting research!! 

Author Response

Please see the detailed point-by-point response and the revised manuscript. Thank you for all your kind efforts. 

Reviewer 3 Report

This is a very interesting and well-conducted study that is novel and has direct implications (theoretical and applied) for various areas of research and practice. I congratulate the authors for the careful design of this study and for presenting the results so clearly. I have few minor comments that I think could improve the overall quality of the paper: 

  1. This is a minor comment, the paper could do with another proofreading as there are some typos and grammar errors throughout the manuscript. 
  2. The authors have acknowledged in the discussion that the sample size is rather small. This is not necessarily a problem, if the study is well-powered to detect reliable associations. As such, I think that the authors could explain briefly how they determined the study's sample size (was this based on the sample size used by previous studies in this area?) and the results of a power analysis to get a sense of the reliability of the results. 
  3. The conclusion is underdeveloped. I think the authors could expand the conclusion to highlight the study's novelty, its main implications and possible directions for future research. I think that most of this information appears already throughout the manuscript but really, I think that the conclusion is where this information should be summarised.  

Author Response

Please see the detailed point-by-point-response and the revised manuscript. Thank you so much for all your kind efforts.
